# ANALOG BITS: GENERATING DISCRETE DATA USING DIFFUSION MODELS WITH SELF-CONDITIONING

**Ting Chen, Ruixiang Zhang[†], Geoffrey Hinton**
Google Research, Brain Team
`{iamtingchen,ruixiangz,geoffhinton}@google.com`

## ABSTRACT

We present *Bit Diffusion*: a simple and generic approach for generating discrete data with continuous state and continuous time diffusion models. The main idea behind our approach is to first represent the discrete data as binary bits, and then train a continuous diffusion model to model these bits as real numbers which we call *analog bits*. To generate samples, the model first generates the analog bits, which are then thresholded to obtain the bits that represent the discrete variables. We further propose two simple techniques, namely Self-Conditioning and Asymmetric Time Intervals, which lead to a significant improvement in sample quality. Despite its simplicity, the proposed approach can achieve strong performance in both discrete image generation and image captioning tasks. For discrete/categorical image generation, we significantly improve previous state-of-the-art on both CIFAR-10 (which has $3K$ discrete 8-bit tokens) and IMAGENET $64\times64$ (which has $12K$ discrete 8-bit tokens), outperforming the best autoregressive model in both sample quality (measured by FID) and efficiency. For image captioning on MS-COCO dataset, our approach achieves competitive results compared to autoregressive models.

## 1 INTRODUCTION

State-of-the-art generative models for discrete data, such as discrete images and text, are based on autoregressive modeling (Van den Oord et al., 2016; Salimans et al., 2017; Parmar et al., 2018; Child et al., 2019; Roy et al., 2021; Jun et al., 2020; Sutskever et al., 2014; Brown et al., 2020; Chowdhery et al., 2022), where the networks, often Transformers (Vaswani et al., 2017), are trained to predict each token given its preceding ones in a sequential manner or with causal attention masks. One major drawback of such approaches is that they typically require computation and memory that is quadratic to the dimension of data (e.g., sequence length or image size), leading to difficulties in modeling large images or sequences. Another drawback is that, during generation, autoregressive models generate one token at a time so the total number of sequential sampling steps is often the same as the dimension of data, making it slow in generating large images or long sequences.

In contrast, diffusion models (Sohl-Dickstein et al., 2015; Ho et al., 2020; Song et al., 2020), or score-based generative models (Song & Ermon, 2019; 2020; Song et al., 2021), can model much higher dimensional data without running into computation and memory issues. During generation, diffusion models iteratively refine samples with a high degree of parallelism, so the total number of sequential sampling steps can be much less than the dimension of data. However, state-of-the-art diffusion models (Dhariwal & Nichol, 2021; Ho et al., 2022; Nichol et al., 2021; Ramesh et al., 2022; Saharia et al., 2022) can only generate continuous data (mainly real valued pixels), and have not yet achieved results competitive with autoregressive models in generating discrete/categorical data, such as generating discrete/categorical images (Hoogeboom et al., 2021; Austin et al., 2021).

In this work, we propose a simple and generic approach for enabling continuous state diffusion models to generate discrete data. The key ingredient in our approach is *analog bits*: real numbers used to model the bits that represent the discrete data. Analog bits can be directly modeled by continuous state diffusion models, without requiring a discrete state space or re-formulation of the continuous

---

[†]Work done as a student researcher at Google.
Code at https://github.com/google-research/pix2seq.

Figure 1: Bit Diffusion: modeling discrete data using continuous diffusion models with analog bits.

diffusion process. At sampling time, the generated analog bits can be decoded into discrete variables by a simple thresholding operation. Our approach, as illustrated in Figure 1, is based on the following high-level conjecture. With strong continuous generative models (diffusion models in particular), it should not be too difficult to generate highly concentrated bimodal data where each real-valued analog bit is close to a binary bit. To reduce the prediction loss (such as negative log likelihood), the network has to model structures among analog bits that can actually lead to meaningful discrete variables after thresholding.

Besides analog bits, we further propose two simple techniques, namely Self-Conditioning and Asymmetric Time Intervals that greatly improve the sample quality. We evaluate the proposed approach on both discrete image generation, and image-conditional text / caption generation. On discrete CIFAR-10 and IMAGENET $64 \times 64$, the proposed Bit Diffusion model significantly improves both existing discrete diffusion models but also the best autoregressive model. For example, on categorical CIFAR-10, the best autoregressive model (Jun et al., 2020) obtains a FID of 12.75, while our model (with $1/3$ of the model size of the autoregressive model, using 100 instead of 3072 sequential inference steps) achieves a much better 6.93. For image captioning on MS-COCO dataset, our model achieves a result competitive with a strong autoregressive captioner based on a Transformer.

## 2 METHOD

**Preliminaries** We start with a short introduction to diffusion models (Sohl-Dickstein et al., 2015; Ho et al., 2020; Song et al., 2020; 2021). Diffusion models learn a series of state transitions to map noise $\epsilon$ from a known prior distribution to $x_0$ from the data distribution. To learn this (reverse) transition from the noise distribution to the data distribution, a forward transition from $x_0$ to $x_t$ is first defined:

$$x_t = \sqrt{\gamma(t)}\, x_0 + \sqrt{1 - \gamma(t)}\, \epsilon, \tag{1}$$

where $\epsilon \sim \mathcal{N}(\mathbf{0}, \mathbf{I})$, $t \sim \mathcal{U}(0, T)$ is a continuous variable, and $\gamma(t)$ is a monotonically decreasing function from 1 to 0. Instead of directly learning a neural net to model the transition from $x_t$ to $x_{t-\Delta}$, one can learn a neural net $f(x_t, t)$ to predict $x_0$ (or $\epsilon$) from $x_t$, and estimate $x_{t-\Delta}$ from $x_t$ and estimated $\tilde{x}_0$ (or $\tilde{\epsilon}$). This training of $f(x_t, t)$ is based on denoising with a $\ell_2$ regression loss:

$$\mathcal{L}_{x_0} = \mathbb{E}_{t \sim \mathcal{U}(0,T), \epsilon \sim \mathcal{N}(\mathbf{0,1})} \| f(\sqrt{\gamma(t)}\, x_0 + \sqrt{1 - \gamma(t)}\, \epsilon, t) - x_0 \|^2. \tag{2}$$

To generate samples from a learned model, it follows a series of (reverse) state transition $x_T \rightarrow x_{T-\Delta} \rightarrow \cdots \rightarrow x_0$. This can be achieved by iteratively applying denoising function $f$ on each state $x_t$ to estimate $x_0$, and then make a transition to $x_{t-\Delta}$ with the estimated $\tilde{x}_0$ (using transition rules such as those specified in DDPM (Ho et al., 2020) or DDIM (Song et al., 2020)). Note that state transitions in these diffusion models assume a continuous data space and state space. Therefore, one cannot directly apply it to model and generate discrete/categorical data.

**Analog Bits** A discrete data variable from an alphabet of size $K$ can be represented using $n = \lceil \log_2 K \rceil$ bits, as $\{0, 1\}^n$. Due to the discreteness, existing work has to re-formulate continuous diffusion models by adopting a discrete data space and state space (Sohl-Dickstein et al., 2015; Hoogeboom et al., 2021; Austin et al., 2021). In contrast, we propose to simply cast the binary bits $\{0, 1\}^n$ into real numbers $\mathbb{R}^n$ for the continuous diffusion models [1]. We term these real numbers *analog bits* since they learn to share the same bimodal values as binary bits but are modeled as real numbers. To draw samples, we follow the same procedure as sampling in a continuous diffusion model, except that we apply a quantization operation at the end by simply thresholding the generated analog bits. This yields binary bits which can be then converted into original discrete/categorical variables. Notably, there is no hard constraint to force the model to generate exact binary bits, but we

---

[1] After casting as real numbers, one may also transform them by shifting and scaling from $0, 1$ to $-b, b$.

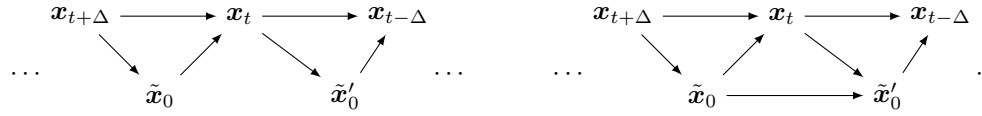

(a) Standard reverse diffusion steps.  (b) Self-Conditioning on the previous $x_0$ estimate.

Figure 2: An illustration of reverse diffusion sampling steps (a) without or (b) with Self-Conditioning. $\tilde{x}_0$ denotes the estimation of data sample by the denoising network $f$ at a sampling step. We propose to condition the network *directly* on its previously generated/estimated samples.

expect a strong continuous generative model to generate real numbers that exhibit very clear bimodal concentrations and this is what happens in our experiments.

For simplicity, we use the same regression loss function (Eq. 2) for modeling analog bits. However, it is possible to use other loss functions such as the cross entropy loss. We also note that the binary encoding mechanism for constructing analog bits is extensible as well (e.g., one-hot encoding). Extensions of loss functions and binary encoding are described in the appendix B.

**Self-Conditioning** Conditioning is a useful technique for improving diffusion models (Nichol & Dhariwal, 2021; Ho et al., 2022). However, a typical conditioning variable is either from some external sources, such as class labels (Nichol & Dhariwal, 2021) or low-resolution images from another network (Nichol & Dhariwal, 2021; Saharia et al., 2021; Ho et al., 2022). Here we propose a technique for the model to directly condition on previously generated samples of its own during the iterative sampling process, which can significantly improve the sample quality of diffusion models.

In a typical diffusion sampling process, the model iteratively predicts $x_0$ (or $\epsilon$) in order to progress the chain of mapping noise into data. However, as shown in Figure 2a, the previously estimated $\tilde{x}_0$ is simply discarded when estimating $x_0$ from a new time step, i.e. the denoising function $f(x_t, t)$ does not directly depend on a previously estimated $\tilde{x}_0$. Here we consider a slightly different denoising function of $f(x_t, \tilde{x}_0, t)$ that also takes previous generated samples as its input, illustrated in Figure 2b. A simple implementation of Self-Conditioning is to concatenate $x_t$ with previously estimated $\tilde{x}_0$. Given that $\tilde{x}_0$ is from the earlier prediction of the model in the sampling chain, this comes at a negligible extra cost during sampling. In order to train the denoising function $f(x_t, \tilde{x}_0, t)$, we make some small changes to the training. With some probability (e.g., $50\%$), we set $\tilde{x}_0 = 0$ which falls back to modeling without Self-Conditioning. At other times, we first estimate $\tilde{x}_0 = f(x_t, 0, t)$ and then use it for Self-Conditioning. Note that we do not backpropagate through the estimated $\tilde{x}_0$ so the overall increase of training time is small (e.g., less than $25\%$).

**Asymmetric Time Intervals** Besides Self-Conditioning, we identify another factor, time step $t$, that can also impact Bit Diffusion models. Time step $t$ is an integral part of both denoising network $f(x_t, t)$ as well as the state transitions. During a typical reverse diffusion process, the model takes symmetric time intervals (i.e., $\Delta$ as in $t \to t - \Delta$) for both the state transition and time reduction itself, resulting in the same/shared $t$ for both arguments of $f(x_t, t)$. However, we find that, when taking large reverse steps, using asymmetric time intervals, implemented via a simple manipulation of time scheduling at generation, can lead to improved sampling quality for Bit Diffusion models.

More specially, with asymmetric time intervals during the sampling process, we have $f(x_t, t')$, where $t' = t + \xi$ and $\xi$ is a small non-negative time difference parameter. Note that training remains unchanged, and the same/shared $t$ is used for both arguments of the $f(x_t, t)$. Figure 3 illustrates the effect with a trained Bit Diffusion model, where it is asked to take two reversing steps from a state $x_t$ constructed using the forward diffusion, and it shows that asymmetric time intervals reduce the number of noisy pixels (after thresholding and converting back to discrete variables).

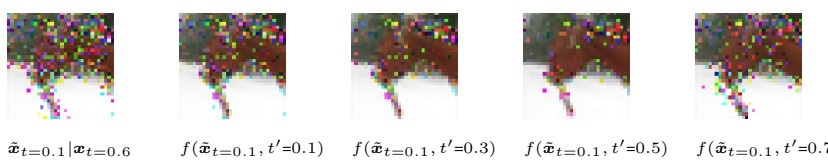

$\tilde{x}_{t=0.1}|x_{t=0.6}$   $f(\tilde{x}_{t=0.1}, t'=0.1)$   $f(\tilde{x}_{t=0.1}, t'=0.3)$   $f(\tilde{x}_{t=0.1}, t'=0.5)$   $f(\tilde{x}_{t=0.1}, t'=0.7)$

Figure 3: When taking a large reverse step from $x_{t=0.6}$ to $x_{t=0.1}$ in Bit Diffusion with maximum time $T = 1.0$, we see that asymmetric time intervals with a positive time difference $\xi$ improve the denoising quality of $x_{t=0.1}$ (by reducing the number of noisy pixels).

**Putting it together** Algorithm 1 and 2 summarize the training and sampling algorithms for the proposed Bit Diffusion model with Analog Bits, Self-Conditioning, and Asymmetric Time Intervals (via the `td` parameter). The proposed changes to the existing diffusion models are highlighted in blue. Note that unlike standard diffusion models (Sohl-Dickstein et al., 2015; Ho et al., 2020; Nichol & Dhariwal, 2021), we use a continuous time parameterization between 0 and 1 instead of a fixed discrete time for maximal flexibility but they perform similarly. More details of the algorithm (including some important functions) can be found in Appendix A.

---

**Algorithm 1** Bit Diffusion training algorithm.

```
def train_loss(x):
  # Binary encoding: discrete to analog bits.
  x_bits = int2bit(x).astype(float)
  x_bits = (x_bits * 2 - 1) * scale

  # Corrupt data.
  t = uniform(0, 1)
  eps = normal(mean=0, std=1)
  x_crpt = sqrt(gamma(t)) * x_bits +
           sqrt(1 - gamma(t)) * eps

  # Compute self-cond estimate.
  x_pred = zeros_like(x_crpt)
  if self_cond and uniform(0, 1) > 0.5:
    x_pred = net(cat([x_crpt, x_pred], -1), t)
    x_pred = stop_gradient(x_pred)

  # Predict and compute loss.
  x_pred = net(cat([x_crpt, x_pred], -1), t)
  loss = (x_pred - x_bits)**2
  return loss.mean()
```

**Algorithm 2** Bit Diffusion sampling algorithm.

```
def generate(steps, td=0):
  x_t = normal(mean=0, std=1)
  x_pred = zeros_like(x_t)

  for step in range(steps):
    # Get time for current and next states.
    t_now = 1 - step / steps
    t_next = max(1 - (step+1+td) / steps, 0)

    # Predict x_0.
    if not self_cond:
      x_pred = zeros_like(x_t)
    x_pred = net(cat([x_t, x_pred], -1), t_now)

    # Estimate x at t_next.
    x_t = ddim_or_ddpm_step(
        x_t, x_pred, t_now, t_next)

  # Binary decoding to discrete data.
  return bit2int(x_pred > 0)
```

---

## 3 EXPERIMENTS

We experiment with two different discrete data generation tasks, namely discrete/categorical image generation, and image captioning (image-conditional text generation).

### 3.1 EXPERIMENTAL SETUP AND IMPLEMENTATION DETAILS

**Datasets** We use CIFAR-10 (Krizhevsky et al., 2009) and IMAGENET $64 \times 64$ (Deng et al., 2009) [2] for image generation experiments. We adopt widely used FID (Heusel et al., 2017) as the main evaluation metric, and it is computed between 50K generated samples and the whole training set. For image captioning, following (Chen et al., 2022), we use MS-COCO 2017 captioning dataset (Lin et al., 2014).

**Binary encoding** Each pixel consists of 3 sub-pixels (RGB channels), and each sub-pixel is an integer in $[0, 256)$ representing the intensity. Standard continuous generative models cast RGB channels as real numbers and normalize them in $[-1, 1]$. For discrete image generation, we consider three discrete encoding for sub-pixels, namely UINT8, GRAY CODE, and UINT8 (RAND). In UINT8, we use 8-bit binary codes converted from the corresponding sub-pixel integer in $[0, 256)$. In GRAY CODE, we assign 8-bit binary codes uniquely to each sub-pixel integer such that two adjacent integers only differ by 1 bit. And in UINT8 (RAND), we assign 8-bit binary codes to every sub-pixel integer by randomly shuffling the integer-to-bits mapping in UINT8. The binary codes in UINT8 and GRAY CODE are loosely correlated with its original sub-pixel intensities, while UINT8 (RAND) has no correlation so each sub-pixel is a categorical variable. The details of the binary codes and their correlations with sub-pixel intensity can be found in the appendix C. We shift and scale the binary bits from $0, 1$ to $-1, 1$ for the analog bits.

For image captioning, we follow (Chen et al., 2022), and use sentencepiece (Kudo & Richardson, 2018) with a vocabulary of size 32K to tokenize the captions. After tokenization, we encode each token into 15 analog bits using the binary codes converted from the corresponding integer. We set the maximum number of tokens to 64 so the total sequence length is 960 bits. Since we directly model bits, it is also possible to directly work with their byte representations without a tokenizer, but we leave this for future work.

---

[2]Following (Brock et al., 2018; Nichol & Dhariwal, 2021), we center crop and area downsample images to $64 \times 64$.

Table 1: Comparison of FIDs on unconditional and class-conditional CIFAR-10. Note that both UINT8 and GRAY CODE are only partial/weakly ordinal (see Appendix C). Our Bit Diffusion achieves state-of-the-art FIDs in generating discrete images, beating the best autoregressive model.

| Method | State space | FID (Unconditional) | FID (Conditional) |
|---|---|---|---|
| *On continuous pixels (as reference):* | | | |
| DDPM (Ho et al., 2020) | Continuous | 3.17 | - |
| DDPM (our reproduction) | Continuous | **3.14** | **2.95** |
| *On discrete (partial) ordinal pixels:* | | | |
| D3PM Gauss+Logistic (Austin et al., 2021) | Discrete | 7.34 | - |
| $\tau$LDR-10 (Campbell et al., 2022) | Discrete | 3.74 | - |
| Bit Diffusion on UINT8 | Continuous | **3.48** | **2.72** |
| Bit Diffusion on GRAY CODE | Continuous | 3.86 | 2.94 |
| *On categorical pixels:* | | | |
| D3PM uniform (Austin et al., 2021) | Discrete | 51.27 | - |
| D3PM absorbing (Austin et al., 2021) | Discrete | 30.97 | - |
| Autoregressive Transformer (Jun et al., 2020) | Discrete | 12.75 | - |
| Bit Diffusion on UINT8 (RAND) | Continuous | **6.93** | **6.43** |

Table 2: Comparison of FIDs on class-conditional IMAGENET 64×64. The corresponding samples can be found in Figure 4 and 11.

| DDPM (our repo.) on continuous pixels | Bit Diffusion on UINT8 | Bit Diffusion on GRAY CODE | Bit Diffusion on UINT8 (RAND) |
|---|---|---|---|
| 3.43 | 4.84 | 5.14 | 8.76 |

**Architecture** We use the U-Net architecture (Ho et al., 2020; Nichol & Dhariwal, 2021; Ronneberger et al., 2015) for image generation. For CIFAR-10, we use a single channel dimension of 256, 3 stages and 3 residual blocks (He et al., 2016) per stage, with a total of 51M parameters. We only use dropout (Srivastava et al., 2014) of 0.3 for continuous diffusion models on CIFAR-10. For IMAGENET 64×64, following (Nichol & Dhariwal, 2021), we use a base channel dimension of 192, multiplied by 1,2,3,4 in 4 stages and 3 residual blocks per stage, which account for a total of 240M parameters [3]. For UINT8 (RAND) encoding, we find the following "*softmax factorization*" architectural tweak on the final output layer can lead to a better performance. Instead of using a linear output layer to predict analog bits directly, we first predict a probability distribution over 256 classes per sub-pixel (with each class corresponds to one of the 256 different 8-bit codes), and then map class distribution into analog bits by taking weighted average over all 256 different 8-bit codes.

For image captioning, we follow the architecture used in (Chen et al., 2021; 2022), with a pre-trained image encoder using the object detection task, for both autoregressive baseline as well as the proposed method. Both decoders are randomly initialized 6-layer Transformer (Vaswani et al., 2017) decoder with 512 dimension per layer. For the autoregressive decoder, the token attention matrix is offset by the causal masks, but it is non-masked all-to-all attention for our Bit Diffusion.

**Other settings** We train our models with the Adam optimizer (Kingma & Ba, 2014). For CIFAR-10, we train the model for 1.5M steps with a constant learning rate of 0.0001 and batch size of 128. For IMAGENET 64×64, we train the model for 500K steps with a constant learning rate of 0.0002 [4] and batch size of 1024. For Bit Diffusion, we use Self-Conditioning by default, unless otherwise specified. We use an exponential moving average of the weights during training with a decay factor of 0.9999. For our best image generation results, we sweep over a few sampling hyper-parameters, such as sampler (DDIM vs DDPM), sampling steps in $\{100, 250, 400, 1000\}$, and time difference in $\{0., 0.01, 0.1, 0.2, 0.5\}$.

## 3.2 DISCRETE IMAGE GENERATION

We compare our model against state-of-the-art generative models (Ho et al., 2020; Austin et al., 2021; Campbell et al., 2022; Jun et al., 2020) on generating discrete CIFAR-10 images in Table 1. Our model

---

[3]Our model is about 30M parameters smaller than that used in (Nichol & Dhariwal, 2021) as we drop the middle blocks for convenience, which may have a minor effect on performance.

[4]For UINT8 (RAND) encoding, we use learning rate of 0.0001 instead.

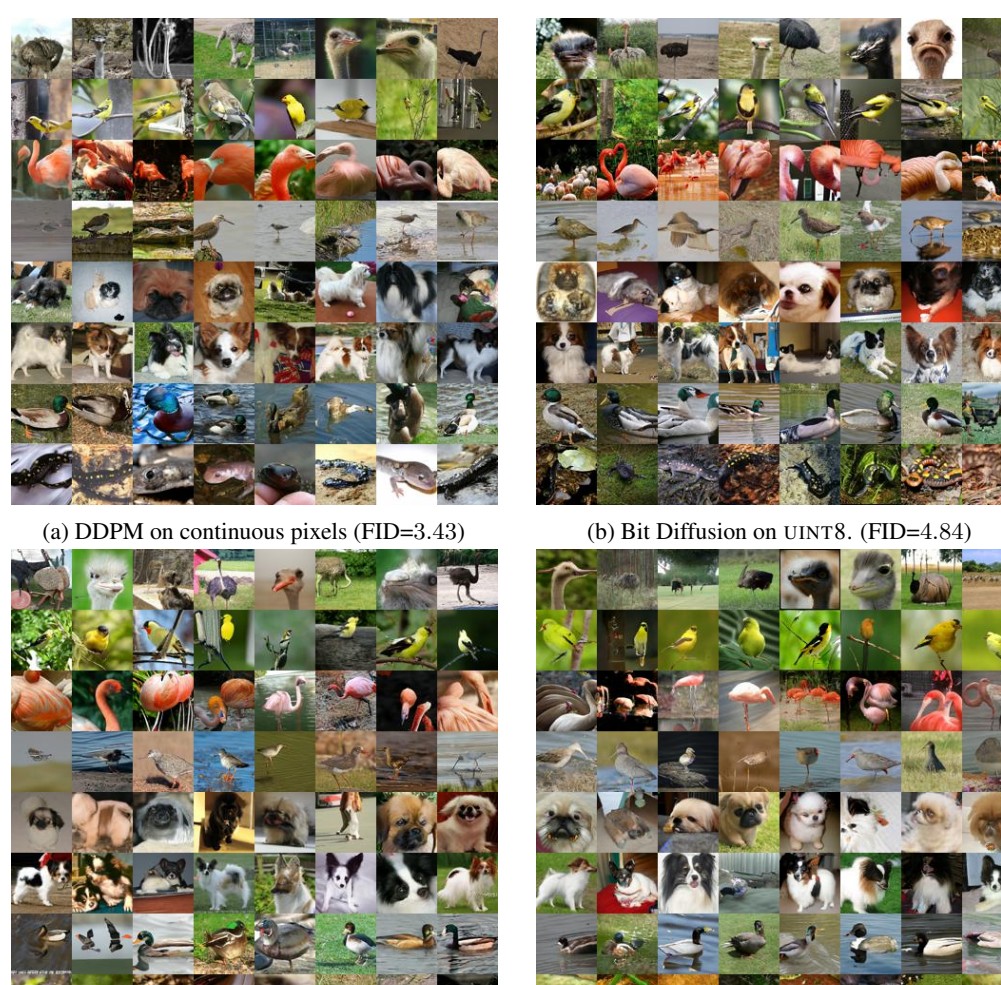

(a) DDPM on continuous pixels (FID=3.43)  (b) Bit Diffusion on UINT8. (FID=4.84)

(c) Bit Diffusion on GRAY CODE (FID=5.14)  (d) Bit Diffusion on UINT8 (RAND) (FID=8.76)

Figure 4: Class-conditional generations on continuous v.s. discrete ImageNet 64×64. Each row represents random samples conditioned on a class, and the classes are adopted from (Nichol & Dhariwal, 2021), namely, 9: ostrich, 11: goldfinch, 130: flamingo, 141: redshank, 154: pekinese, 157: papillon, 97: drake and 28: spotted salamander. More samples from random classes are shown in Figure 11.

achieves better results compared to both existing discrete diffusion models and the best autoregressive model. When compared to continuous diffusion models (i.e., DDPM), our Bit Diffusion models on UINT8 and GRAY CODE can achieve similar performance.

Discrete generation of IMAGENET 64×64 is significantly harder than CIFAR-10, and we have not found other competing methods that report FIDs, so we only compare the proposed method against DDPM on continuous pixels. Results are shown in Table 2. We find that the diffusion model on continuous pixels has the best FID while the diffusion model on UINT8 (RAND), i.e., categorical data, has the worst FID, indicating the increase of hardness when removing intensity/order information in sub-pixels. Note that, in these experiments, there is no extra model capacity to compensate for the loss of intensity/order information since the model sizes are the same. Figure 4 shows generated images of different diffusion models on continuous and discrete IMAGENET 64×64. Despite the differences in FIDs, visually these samples look similar.

**Ablation of Self-Conditioning** Figure 5 shows the effectiveness of the Self-Conditioning technique in both Bit Diffusion and continuous diffusion models. Note that the experiments are performed in three settings, namely CIFAR-10 with UINT8, CIFAR-10 with UINT8 (RAND), and IMAGENET

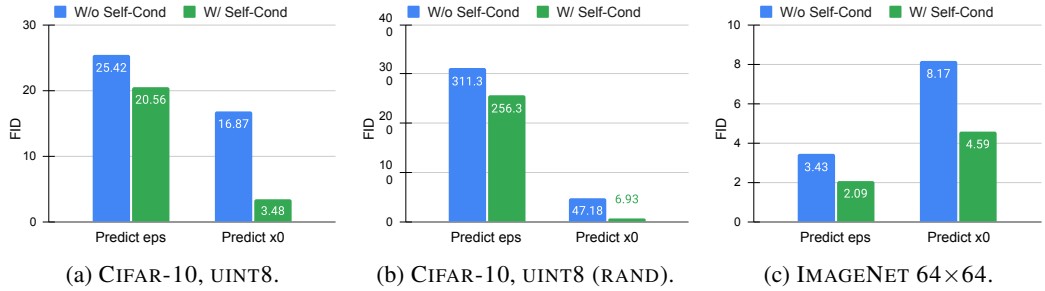

(a) CIFAR-10, UINT8.  (b) CIFAR-10, UINT8 (RAND).  (c) IMAGENET 64×64.

Figure 5: Self-conditioning is a generic technique that not only greatly improves Bit Diffusion but also leads to improved results for continuous diffusion models.

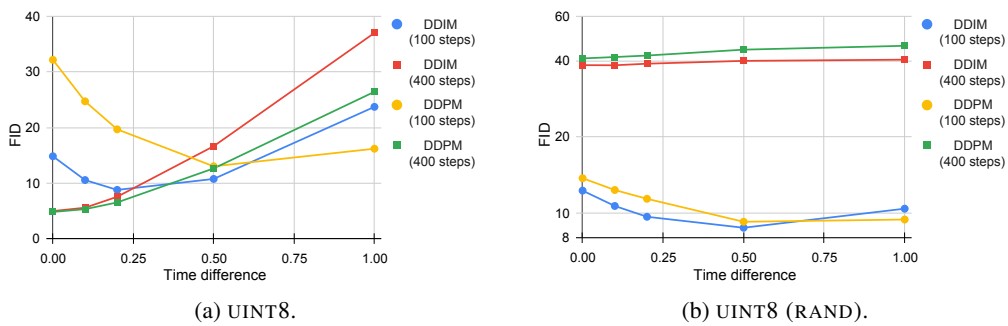

(a) UINT8.  (b) UINT8 (RAND).

Figure 6: Effect of time difference in class-conditional IMAGENET 64×64. Optimal time difference shrinks to zero as the number of sampling steps increases. For 100 sampling steps, non-zero time difference leads to improved FIDs.

64×64 with continuous pixels, where the only difference for pairs in each setting is whether the Self-Conditioning is used. For CIFAR-10, we find that Self-Conditioning greatly improves the performance across different binary encodings. We also notice that for Bit Diffusion, predicting $x_0$ is much more effective than predicting $\epsilon$. For IMAGENET 64×64, we find that the proposed Self-Conditioning also leads to improved FIDs for continuous diffusion (i.e., DDPM). Therefore, we conclude that Self-Conditioning by itself is a generic technique that can benefit diffusion models on both continuous and discrete data.

**Ablation of asymmetric time intervals** Figure 6 shows the FID on generated IMAGENET 64×64 samples as we vary the time difference parameter during the sampling process. We find that as the number of steps increases (from 100 to 400), the optimal time difference shrinks to 0. For 100 steps, a non-zero time difference leads to a significant improvement of FID. We also note that for Bit Diffusion on UINT8 (RAND), using 400 sampling steps actually leads to a drastically worse sample quality than using 100 steps. This is related to how the Self-Conditioning is applied and we present alternative Self-Conditioning sampling strategies in the Appendix G, some of which lead to improved FIDs at a cost of longer sampling time.

**Concentration of generated analog bits** Figure 7 visualizes the distribution of generated analog bits from 64 generated images on IMAGENET 64×64. Although there is no hard constraint on the analog bits being binary / bimodal, the generated ones are highly concentrated on two modes, which makes the thresholding / quantization easy and robust.

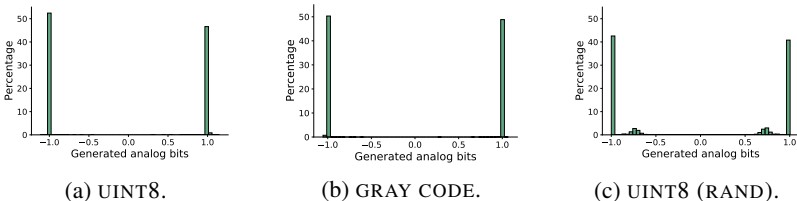

(a) UINT8.  (b) GRAY CODE.  (c) UINT8 (RAND).

Figure 7: Histogram distribution (50 bins) of the analog bits from 64 randomly generated IMAGENET 64×64 samples at $\tilde{x}_0$, with 100 DDIM steps. Most of the generated analog bits are very concentrated.

### 3.3 IMAGE CAPTIONING

We compare our Bit Diffusion model with an autoregressive Transformer baseline (Chen et al., 2022). As mentioned, both models have similar architectures, with an object detection pretrained (Chen et al., 2021) image encoder, and a randomly initialized Transformer (Vaswani et al., 2017) decoder. Table 3 presents the main comparison. Overall, our model achieves similar performance as the autoregressive model. We find that generally it only needs about 10 steps for the model to achieve good results, despite that there are a total of maximum 960 bits for caption that the model has to model. We find that the asymmetric time intervals play an important role in the final performance of our model, as demonstrated in Table 4, especially when sampling steps are fewer.

Table 3: Image captioning results on MS-COCO dataset with a randomly initialized text decoder.

| Method | BLEU-4 | CIDEr | ROUGE-L |
|---|---|---|---|
| Autoregressive Transformer | 33.9 | **1.18** | 0.57 |
| Bit Diffusion (5 steps) | 31.5 | 1.00 | 0.55 |
| Bit Diffusion (10 steps) | 34.5 | 1.13 | 0.57 |
| Bit Diffusion (20 steps) | **34.7** | 1.15 | **0.58** |
| Bit Diffusion (40 steps) | 34.4 | 1.15 | 0.57 |

Table 4: Asymmetric time intervals significantly improves the performance of Bit Diffusion.

| | Time difference | | | | | | | | |
|---|---|---|---|---|---|---|---|---|---|
| | 0.0 | 1.0 | 2.0 | 3.0 | 4.0 | 5.0 | 6.0 | 7.0 | 8.0 |
| 5 steps | 17.1 | 27.8 | 30.8 | 31.5 | **31.6** | 31.5 | 31.5 | 31.5 | **31.6** |
| 10 steps | 17.6 | 26.3 | 30.7 | 32.6 | 33.4 | 34.0 | 34.3 | 34.5 | **34.6** |
| 20 steps | 20.0 | 27.9 | 30.6 | 32.0 | 32.3 | 33.9 | 34.4 | **34.7** | 34.5 |
| 40 steps | 20.7 | 27.5 | 30.7 | 32.2 | 32.9 | 33.2 | 33.8 | **34.4** | **34.4** |

Table 5 provides some generated samples of our model when different inference steps are used. The model makes mistakes when the sampling steps are too few, and the mistakes may not always be interpretable due to that the model directly predicts the bits behind the tokenized word pieces and a small difference in bits can lead to total different words.

Table 5: Generated image captions under different number of sampling steps.

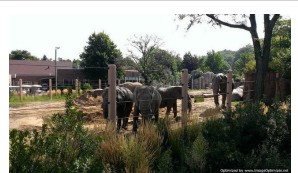

Steps=1: A group of diets in for foraa

Steps=2: A group of elephants in in a\ufffd.

Steps=5: A group of elephants standing in a Irish.

Steps=10: A group of elephants standing by a fence.

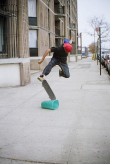

Steps=1: A says man fora You a\u0000.

Steps=2: A says man a skateboard' aa.

Steps=5: A man on a skateboard on a skateboard.

Steps=10: A man is doing a trick on a skateboard.

## 4 RELATED WORK

**Autoregressive models for discrete data** Autoregressive models have demonstrated state-of-the-art results when it comes to generating discrete data. In particular, text generation, or language modeling, is dominated by autoregressive approaches (Sutskever et al., 2014; Brown et al., 2020; Chowdhery et al., 2022). Autoregressive models are also applied to discrete/categorical image generation (Van den Oord et al., 2016; Salimans et al., 2017; Parmar et al., 2018; Child et al., 2019; Roy et al., 2021; Jun et al., 2020; Chen et al., 2020a), where they work well on small image resolutions. However, the computation cost and memory requirement increase drastically (typically in a quadratic relation) as

the size of sequence or the image resolution increase, so it becomes very challenging to scale these approaches to data with large dimensions.

**Diffusion models for discrete data** State-of-the-art diffusion models (Dhariwal & Nichol, 2021; Ho et al., 2022; Nichol et al., 2021; Ramesh et al., 2022; Saharia et al., 2022) cannot generate discrete or categorical data. Existing extensions of these continuous diffusion models to discrete data are based on both discrete data space and state space (Sohl-Dickstein et al., 2015; Hoogeboom et al., 2021; Austin et al., 2021; Campbell et al., 2022). Compared to discrete state space, continuous state space is more flexible and potentially more efficient. Our approach is also compatible with both discrete and continuous time, and does not require re-formulation of existing continuous models, thus it is simpler and can potentially be plugged into a broader family of generative models.

Another line of discrete diffusion models is based on the embedding of discrete data (Li et al., 2022). One can also consider our binary encoding with analog bits as a simple fixed encoder, and the decoding / quantization of bimodal analog bits is easy and robust via a simple thresholding operation. In contrast, the quantization of real numbers in generated continuous embedding vectors may contain multiple modes per dimension, leading to potential difficulty in thresholding/quantization.

**Normalizing Flows for discrete data** Normalizing Flows (Rezende & Mohamed, 2015; Dinh et al., 2017; Kingma & Dhariwal, 2018) are a powerful family of generative models for high-dimensional continuous distributions based on some invertible mapping. However, straightforward application of flow-based models on categorical data is limited due to the inherent challenges on discrete support. Discrete flows (Tran et al., 2019; Hoogeboom et al., 2019; Lindt & Hoogeboom, 2021) introduce invertible transformations of random variables in discrete space without the need of computing the log-determinant of Jacobian. Other works (Lippe & Gavves, 2021; Hoogeboom et al., 2021; Tan et al., 2021) introduce various embedding methods for transforming discrete data into continuous space with disjoint support, which can be interpreted as a variational inference problem (Theis et al., 2015) with different dequantization distribution families. Several works (Kingma et al., 2016; Ziegler & Rush, 2019; Zhang et al., 2020) also explore normalizing flows on discrete data under the Variational Autoencoders (Kingma & Welling, 2013) framework by enriching the prior. Compared to our diffusion-based approach, these models suffer from strict invertible restrictions on network architecture, thus limiting their capacity.

**Other generative models for discrete data** Other generative models, such as Varational Autoencoders (VAE) (Kingma & Welling, 2013), Generateive Adversarial Networks (GAN) (Goodfellow et al., 2014; Yu et al., 2017; Che et al., 2017; Hjelm et al., 2017; Fedus et al., 2018) have also been applied to generate discrete data. These methods have not yet achieved the level of performance as autoregressive models on tasks such as discrete image generation or text generation, in terms of sample quality or data likelihood. Potentially, the proposed analog bits can also be applied to these continuous generative models, by having the networks directly model and generate analog bits, but it is not explored in this work.

**Other related work** The proposed Self-Conditioning technique shares some similarities with self-modulation in GANs (Chen et al., 2018a) (where the earlier latent state can directly modulate the later latent states) and SUNDAE (Savinov et al., 2021) (where an inference step is incorporated for denoising).

## 5 CONCLUSION

We introduce a simple and generic technique that enables continuous state diffusion models to generate discrete data. The main idea is to encode discrete or categorical data into bits and then model these bits as real numbers that we call analog bits. We also propose two simple techniques, namely Self-Conditioning (i.e., condition the diffusion models directly on their previously generated samples) and Asymmetric Time Intervals, that lead to improved sample quality. We demonstrate that our approach leads to state-of-the-art results in discrete / categorical image generation, beating the best autoregressive model. In an image-conditional text generation task on MS-COCO dataset, we also achieve competitive results compared to autoregressive models. One limitation of our approach, similar to other existing diffusion models, is that they still require a significant number of inference steps for generating good (image) samples. However, we expect that future improvements from diffusion models for continuous data can also transfer to discrete data using analog bits.

## ACKNOWLEDGEMENTS

We would like to thank Priyank Jaini, Kevin Swersky for providing helpful feedback to our draft. Our implementation is partially based on the Pix2Seq codebase, and we thank Lala Li, Saurabh Saxena, for their contributions to the Pix2Seq codebase.

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

## A   More Details of Algorithm 1 and 2

Algorithm 3 and 4 provide more detailed implementations of functions in Algorithm 1 and 2.

---
**Algorithm 3** Binary encoding and decoding algorithms (in Tensorflow).

---

```python
import tensorflow as tf

def int2bit(x, n=8):
  # Convert integers into the corresponding binary bits.
  x = tf.bitwise.right_shift(tf.expand_dims(x, -1), tf.range(n))
  x = tf.math.mod(x, 2)
  return x

def bit2int(x):
  # Convert binary bits into the corresponding integers.
  x = tf.cast(x, tf.int32)
  n = x.shape[-1]
  x = tf.math.reduce_sum(x * (2 ** tf.range(n)), -1)
  return x
```

---

---
**Algorithm 4** $x_t$ estimation with DDIM / DDPM updating rules.

---

```python
def gamma(t, ns=0.0002, ds=0.00025):
  # A scheduling function based on cosine function.
  return numpy.cos(((t + ns) / (1 + ds)) * numpy.pi / 2)**2

def ddim_step(x_t, x_pred, t_now, t_next):
  # Estimate x at t_next with DDIM updating rule.
  γnow = gamma(t_now)
  γnext = gamma(t_next)
  x_pred = clip(x_pred, -scale, scale)
  eps = 1/√(1-γnow) * (x_t - √γnow * x_pred)
  x_next = √γnext * x_pred + √(1 - γnext) * eps
  return x_next

def ddpm_step(x_t, x_pred, t_now, t_next):
  # Estimate x at t_next with DDPM updating rule.
  γnow = gamma(t_now)
  αnow = gamma(t_now) / gamma(t_next)
  σnow = sqrt(1 - αnow)
  z = normal(mean=0, std=1)
  x_pred = clip(x_pred, -scale, scale)
  eps = 1/√(1-γnow) * (x_t - √γnow * x_pred)
  x_next = 1/√αnow * (x_t - (1-αnow)/√(1-γnow) * eps) + σnow * z
  return x_next
```

---

## B   Alternative Binary Encoding and Loss Functions

### B.1   Analog Bits based on One-Hot Encoding

An alternative binary encoding to the base-2 encoding of the discrete data used in the main paper, is the one-hot encoding, where a discrete variable is represented as a vector whose length is the same as the vocabulary size $K$, with a single slot being $1$ and the rest being $0$. The resulting one-hot vector can be similarly treated as analog bits and modeled by continuous state diffusion models. To obtain discrete variables corresponding to the generated analog bits, we use an $\arg\max$ operation over all candidate categories, instead of the thresholding operation in base-2 analog bits. Note that the one-hot encoding requires $K$ bits, which is less efficient compared to base-2 encoding that only requires $\lceil \log_2 K \rceil$ bits, especially for large $K$. [5]

---
[5]Although, one can reduce the vocabulary size $K$ by using sub-tokenization (e.g., subword (Sennrich et al., 2015)), or learned discrete codes (Chen et al., 2018b; 2020b).

### B.2 SIGMOID CROSS ENTROPY LOSS

As we use $\ell_2$ loss by default for its simplicity and compatibility with continuous diffusion models. The proposed Bit Diffusion models can work with other loss functions too. Since the analog bits are bimodal, we can use the following sigmoid cross entropy loss:

$$\mathcal{L}_{\boldsymbol{x}_0, \boldsymbol{x}_t, t} = \log \sigma(\boldsymbol{x}_0 f(\boldsymbol{x}_t, t)),$$

where we assume $\boldsymbol{x}_0 \in \{-1, 1\}^n$, and $\sigma$ is a sigmoid function. During the sampling process, we use $2\sigma(f(\boldsymbol{x}_t, t)) - 1$ as the output of denoising network.

### B.3 SOFTMAX CROSS ENTROPY LOSS

For one-hot analog bits, one could also add a softmax activation function for the output of denosing network $f$, and use the following softmax cross entropy loss:

$$\mathcal{L}_{\boldsymbol{x}_0, \boldsymbol{x}_t, t} = \boldsymbol{x}_0 \log \text{softmax}(f(\boldsymbol{x}_t, t)),$$

where we assume $\boldsymbol{x}_0 \in \{0, 1\}^n$ which is the one-hot representation.

### B.4 PRELIMINARY EXPERIMENTS

Table 6 presents FIDs of Bit Diffusion models with different types of analog bits and loss functions on unconditional CIFAR-10. Note that it is possible some of these results can be improved by more tuning of hyper-parameters or tweaks of the network, but we do not focus on them in this work.

Table 6: FIDs of Bit Diffusion models with different types of analog bits and loss functions on unconditional CIFAR-10.

|  | $\ell_2$ loss | Logistic loss | Softmax loss |
|---|---|---|---|
| ONE HOT | 46.32 | 26.82 | 29.49 |
| UINT8 | 3.48 | 3.53 | - |
| GRAY CODE | 3.86 | 3.71 | - |
| UINT8 (RAND) | 6.93 | 49.29 | - |

## C ON BINARY ENCODING OF PIXELS: UINT8, GRAY CODE, UINT8 (RAND)

In the main paper, we describe three different types of binary encodings of pixels. Here we provide additional detail on how we generate UINT8 (RAND): we first apply a random permutation to 256 sub-pixel values, and then assign the binary binary bits of permuted integers to the non-permuted integers. For example, assume 0 is mapped to 228 after the permutation, the analog bits of 0 would be the binary bits of 228. The random permutation is generated by `numpy.random.seed(42);` `numpy.random.shuffle(numpy.arange(256))`.

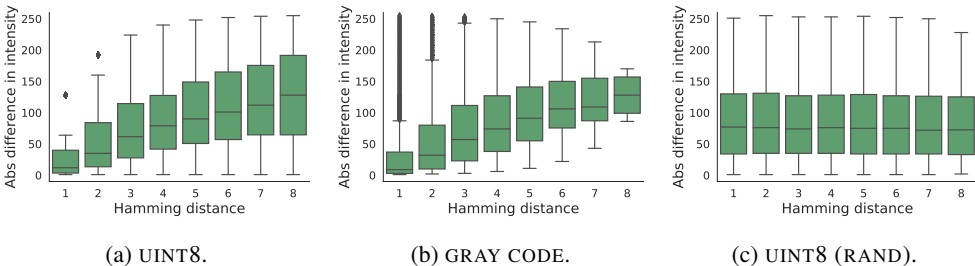

(a) UINT8.  (b) GRAY CODE.  (c) UINT8 (RAND).

Figure 8: Correlation between (absolute) difference in subpixel intensity and the Hamming distance of the corresponding binary bits.

Figure 8 show the correlation between Hamming distance of three different binary encodings we use and the (absolute) difference of sub-pixel intensity. This is done by taking every pair of subpixel

integers (in $[0, 256)$), compute their absolute difference, as well as the Hamming distance between the corresponding binary bits. We find that both UINT8 and GRAY CODE exhibit partial correlation between the two quantities (with different correlation patterns), meaning that these codes partially contain the order information about the original sub-pixel intensity. However, UINT8 (RAND) exhibits no correlation between hamming distance and sub-pixel intensity, indicating the order information is fully removed, thus can be considered as categorical data.

## D    A TOY EXAMPLE ON CONTINUOUS MODELING OF DISCRETE VARIABLES

An intuitive toy example of how a continuous generative model can generate binary data is given in Figure 9, where a mapping from prior distribution at $x_T$ to data distribution at $x_0$ is shown. With a deterministic sampler (such as DDIM), it is straight-forward how they can represent any Bernoulli distribution by dividing the prior into two regions of probability densities corresponding to the Bernoulli distribution. For stochastic samplers, they can achieve a similar effect but the mapping from noise to data is stochastic. For an arbitrary discrete variable, represented as m-dimensional Bernoulli distribution, the mapping from continuous noise distribution to the target Bernoulli distribution also exists but it is more complicated (and difficult to visualize).

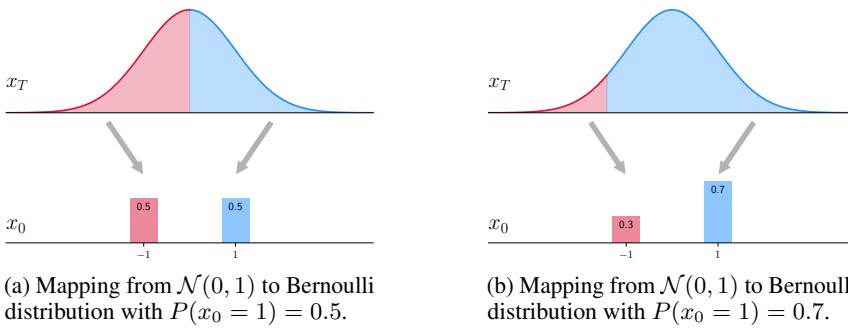

(a) Mapping from $\mathcal{N}(0, 1)$ to Bernoulli distribution with $P(x_0 = 1) = 0.5$.

(b) Mapping from $\mathcal{N}(0, 1)$ to Bernoulli distribution with $P(x_0 = 1) = 0.7$.

Figure 9:  A toy example on continuous modeling of discrete variables.

## E    ON OTHER SAMPLERS FOR CONTINUOUS DIFFUSION MODELS

As our models are based on continuous diffusion models, in theory our models are able to incorporate faster samplers. To this end, we conduct preliminary exploration of using DPM-Solver (Lu et al., 2022) for sampling some of our models.

We find that DPM-Solver provides a boost to diffusion models based on analog bits, similar to what it is able to do for continuous data. This shows a potential of our model enjoying faster continuous sampler while other baselines (e.g., D3PM) may not be able to do due to their use of discrete states. Table 7 below shows the FID scores of bit diffusion models on ImageNet-64x64 under different binary encoding schemes. We find that the DPM-Solver is able to provide a significant reduction in function evaluations for bit diffusion on discrete/categorical data (with 30 NFEs it gets comparable FIDs as 100 NFEs of DDIM), similar to that in continuous diffusion models.

Furthermore, we also find that self-conditioning continues to provide a boost with DPM-solver. For example, the table 8 shows FID scores of diffusion models on ImageNet 64x64 (continuous rgb values). And we find that the self-conditioning consistently improves the performance of DPM-Solver with fixed number of function evaluations.

## F    EXTRA RANDOM SAMPLES ON CIFAR-10 AND IMAGENET $64 \times 64$

Figure 10 shows random samples (non cherry-picked) from unconditional diffusion models on CIFAR-10 with continuous pixels and analog bits.

Figure 11 shows random samples (non cherry-picked) from class-conditional diffusion models on IMAGENET $64 \times 64$ with continuous pixels and analog bits.

Table 7: Comparison of Continuous Diffusion Samplers. FIDs on ImageNet 64×64 shown below.

| Samplers | UINT8 | Gray Code | UINT8 (RAND) |
|---|---|---|---|
| DDIM @ 1000 NFE | 5.51 | 5.14 | 58.45 |
| DDPM @ 1000 NFE | 7.71 | 6.91 | 64.08 |
| DDIM @ 400 NFE | 5.00 | 5.52 | 38.44 |
| DDPM @ 400 NFE | 4.84 | 5.37 | 40.91 |
| DDIM @ 100 NFE | 8.80 | 11.31 | 8.76 |
| DDPM @ 100 NFE | 13.04 | 12.77 | 9.25 |
| DPM-Solver @ 30 NFE | 7.85 | 9.64 | 10.39 |
| DPM-Solver @ 50 NFE | 6.46 | 7.61 | 10.96 |

Table 8: The effect of Self-Conditioning for sampling with DPM-Solver. FIDs on ImageNet 64×64 shown below.

| Model | DPM-Solver @ 20 NFE | DPM-Solver @ 30 NFE |
|---|---|---|
| $\epsilon$ prediction, w/o self-conditioning | 6.10 | 5.58 |
| $\epsilon$ prediction, w/ self-conditioning | 4.24 | 4.15 |
| $x_0$ prediction, w/o self-conditioning | 12.13 | 11.05 |
| $x_0$ prediction, w/ self-conditioning | 6.94 | 6.43 |

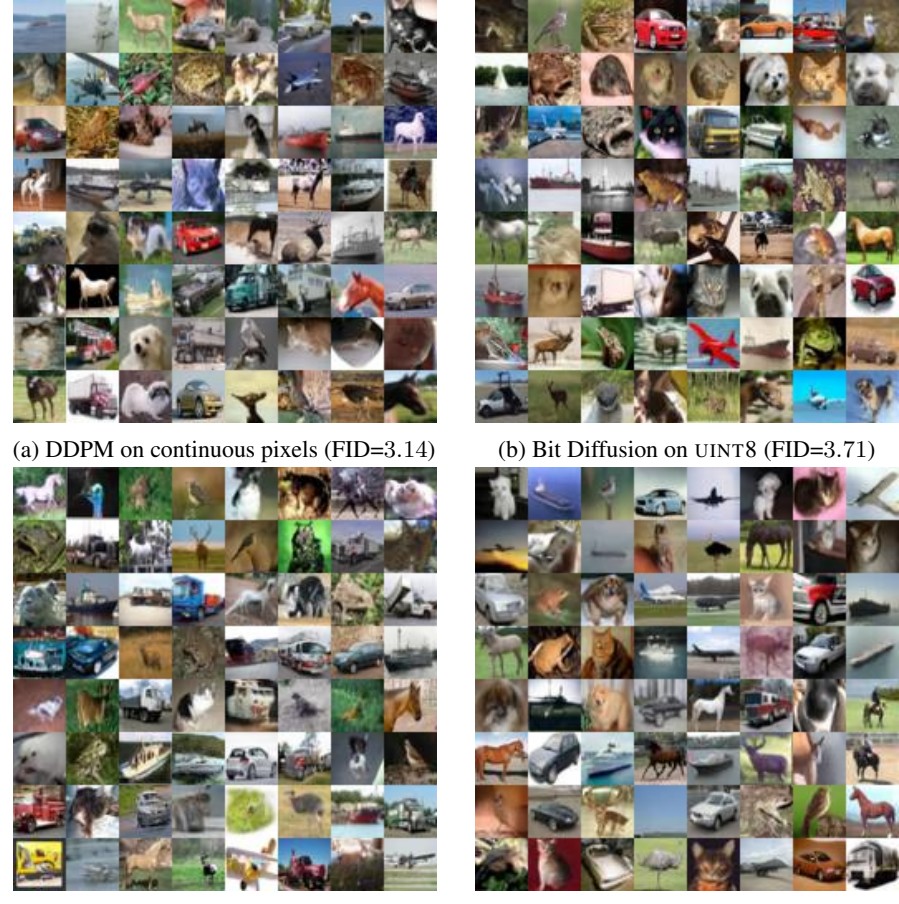

(a) DDPM on continuous pixels (FID=3.14)  (b) Bit Diffusion on UINT8 (FID=3.71)

(c) Bit Diffusion on GRAY CODE (FID=3.88)  (d) Bit Diff. on UINT8 (RAND) (FID=6.93)

Figure 10: Random samples from unconditional models trained on CIFAR-10. (a) is for continuous image generation, (b), (c), and (d) are for discrete/categorial image generation.

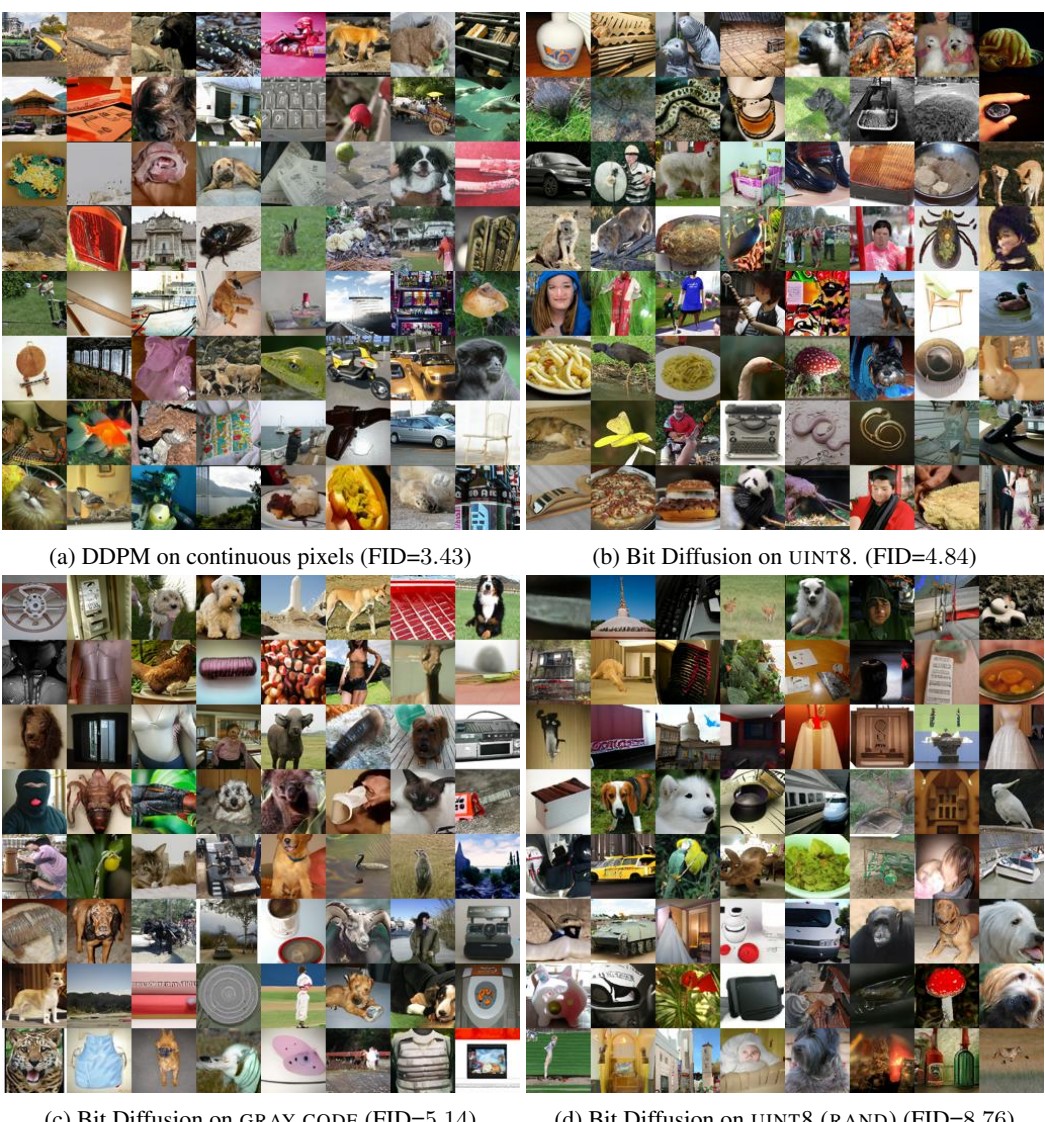

(a) DDPM on continuous pixels (FID=3.43)          (b) Bit Diffusion on UINT8. (FID=4.84)

(c) Bit Diffusion on GRAY CODE (FID=5.14)          (d) Bit Diffusion on UINT8 (RAND) (FID=8.76)

Figure 11: Random samples from class-conditional models trained on IMAGENET 64×64. (a) is for continuous image generation, (b), (c), and (d) are for discrete/categorial image generation.

# G  ON SAMPLING STRATEGIES WITH SELF-CONDITIONING

## G.1  METHOD

In this section, we present extensions to the default sampling strategy with Self-Conditioning in Algorithm 2. The default sampling strategy utilizes data estimate from the previous step as the conditional input to the denoising network for producing data estimate at the current step. While this is both simple and effective, we observe that, for UINT8 (RAND) encoding of pixels, as the number of sampling steps increases (with both DDIM or DDPM samplers), the generated samples tend to be over-smoothed. We propose the following two extensions of the default sampling strategy to mitigate the issues and provide improvements when using larger sampling steps.

**Self-Conditioning based on Momentum Estimate**  The first extension to the default sampling strategy is to adopt an exponential moving average over the previous data estimate to provide a more reliable conditioning input, similar to a momentum optimizer. The detailed procedure is shown in algorithm 5, where the differences from the default sampling strategy are highlighted in blue. Note that the default sampling strategy can also be considered as a special case of this generalized form in that the momentum is set to zero.

**Self-Conditioning based on Self-Guidance**  One potential issue with the default sampling strategy is the slight discrepancy of the Self-Conditioning signal during training and inference/sampling. Specifically, during training, the Self-Conditioning signal is the data estimate from the same time step, while, during sampling, it is from the past time step(s). Therefore, here we propose an approach that also use the same step data estimate for self-conditioning, which comes at the cost of extra forward pass over the denoising network at sampling time. Specifically, we conduct two forward passes of denoising network per sampling step, one with zero data estimate and the other with current data step estimate, and then we use a weighted combination, similar to (Ho & Salimans, 2021), of both prediction to form the final prediction at the current step. The detailed procedure is given in algorithm 6 with differences to the default sampling strategy highlighted.

---

**Algorithm 5** Sampling with Self-Conditioning based on Momentum Estimate.

```
def generate(steps, td=0, momentum=0.):
 x_t = normal(mean=0, std=1)
 x_pred = zeros_like(x_t)
 x_accu = zeros_like(x_t)

 for step in range(steps):
   # Get time for current and next states.
   t_now = 1 - step / steps
   t_next = max(1 - (step + 1 + td) / steps, 0)

   # Predict x_0 (with self-cond).
   x_accu = momentum * x_accu + (1 - momentum) * x_pred
   x_pred = net(cat([x_t, x_accu], -1), t_now)

   # Estimate x at t_next.
   x_t = ddim_or_ddpm_step(x_t, x_pred, t_now, t_next)

 # Binary decoding: analog bits to discrete data.
 x_int = bit2int(x_pred > 0)
 return x_int
```

---

---

**Algorithm 6** Sampling with Self-Conditioning based on Self-Guidance.

---

```
def generate(steps, td=0, guide_w=3.0):
 x_t = normal(mean=0, std=1)
 x_pred = zeros_like(x_t)

 for step in range(steps):
   # Get time for current and next states.
   t_now = 1 - step / steps
   t_next = max(1 - (step + 1 + td) / steps, 0)

   # Predict x_0 wo/ self-cond.
   x_pred_uncond = net(cat([x_t, zeros_like(x_t)], -1), t_now)
   # Predict x_0 w/ self-cond.
   x_pred_selfcond = net(cat([x_t, x_pred_uncond], -1), t_now)
   # Apply self-guidance.
   x_pred = guide_w * x_pred_selfcond + (1.0 - guide_w) * x_pred_uncond

   # Estimate x at t_next.
   x_t = ddim_or_ddpm_step(x_t, x_pred, t_now, t_next)

 # Binary decoding: analog bits to discrete data.
 x_int = bit2int(x_pred > 0)
 return x_int
```

---

## G.2 EXPERIMENTS

Table 9 reports the best FID scores across various sampling strategies discussed here (as well as samplers, sampling steps, time difference in asymmetric time intervals).

Table 9: Best FIDs of Bit Diffusion models with different Self-Conditioning sampling strategies on conditional IMAGENET $64\times64$.

| | UINT8 | GRAY CODE | UINT8 (RAND) |
|---|---|---|---|
| Default sampling (momentum= 0) | **4.84** | **5.14** | 8.76 |
| Momentum Estimate | **4.85** | **5.14** | 8.51 |
| Self-Guidance | 5.15 | 5.65 | **7.87** |

Figure 12 shows FIDs on conditional IMAGENET $64\times64$ with UINT8 encoding, using Momentum Estimate with different sampling steps. We find that the momentum on the data estimate is only helpful when sampling steps are larger.

Figure 13 shows FIDs on conditional IMAGENET $64\times64$ with UINT8 encoding, using Self-Guidance with different sampling steps. We find that a guidance weight between 3.0 and 5.0 is generally preferable and robust to other hyper-parameters (such as sampler choice, sampling steps, and time difference).

## G.3 SAMPLES

Figure 14 and 15 provide generated samples from different sampling strategies with 100 and 1000 DDIM sampling steps, respectively.

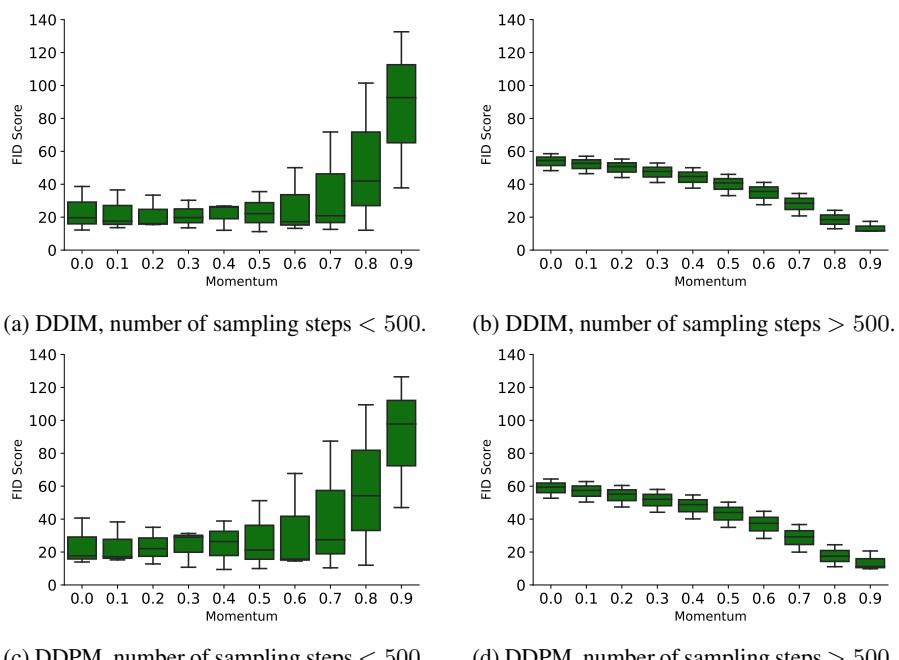

(a) DDIM, number of sampling steps $< 500$. (b) DDIM, number of sampling steps $> 500$.

(c) DDPM, number of sampling steps $< 500$. (d) DDPM, number of sampling steps $> 500$.

Figure 12: FID on conditional IMAGENET $64\times64$ with UINT8 (RAND) encoding using self-condition sampling based on momentum estimate. The statistics of FID scores in each group are aggregated over the number of sampling steps in $\{100, 200, 400, 600, 800, 1000\}$, time difference in $\{0.0, 0.2, 0.4, 0.6, 0.8\}$.

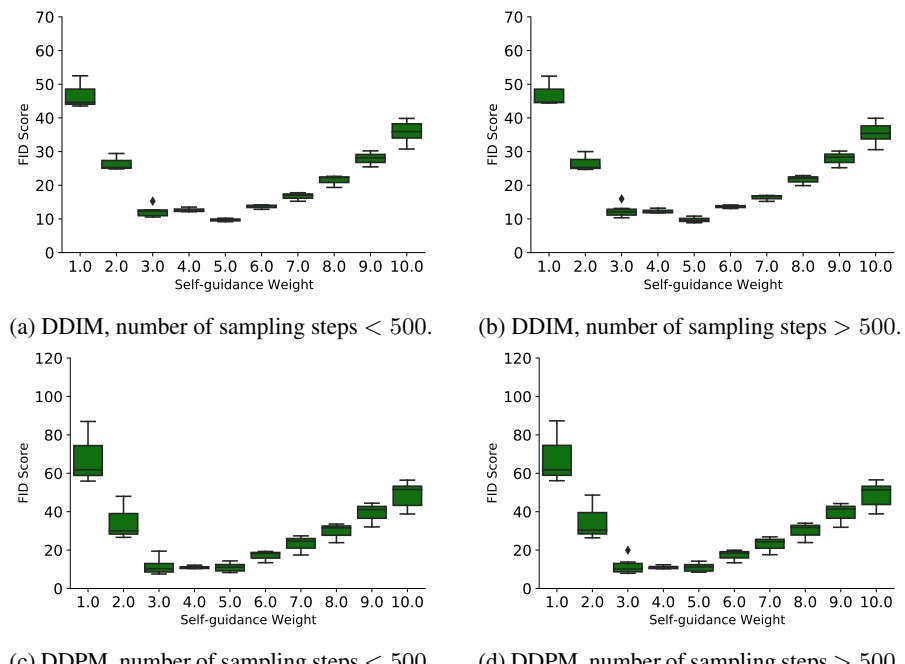

(a) DDIM, number of sampling steps $< 500$. (b) DDIM, number of sampling steps $> 500$.

(c) DDPM, number of sampling steps $< 500$. (d) DDPM, number of sampling steps $> 500$.

Figure 13: FID on conditional IMAGENET $64\times64$ with UINT8 (RAND) encoding using self-condition sampling based on self-guidance. The statistics of FID scores in each group are aggregated over the number of sampling steps in $\{100, 200, 400, 600, 800, 1000\}$, time difference in $\{0.0, 0.1\}$.

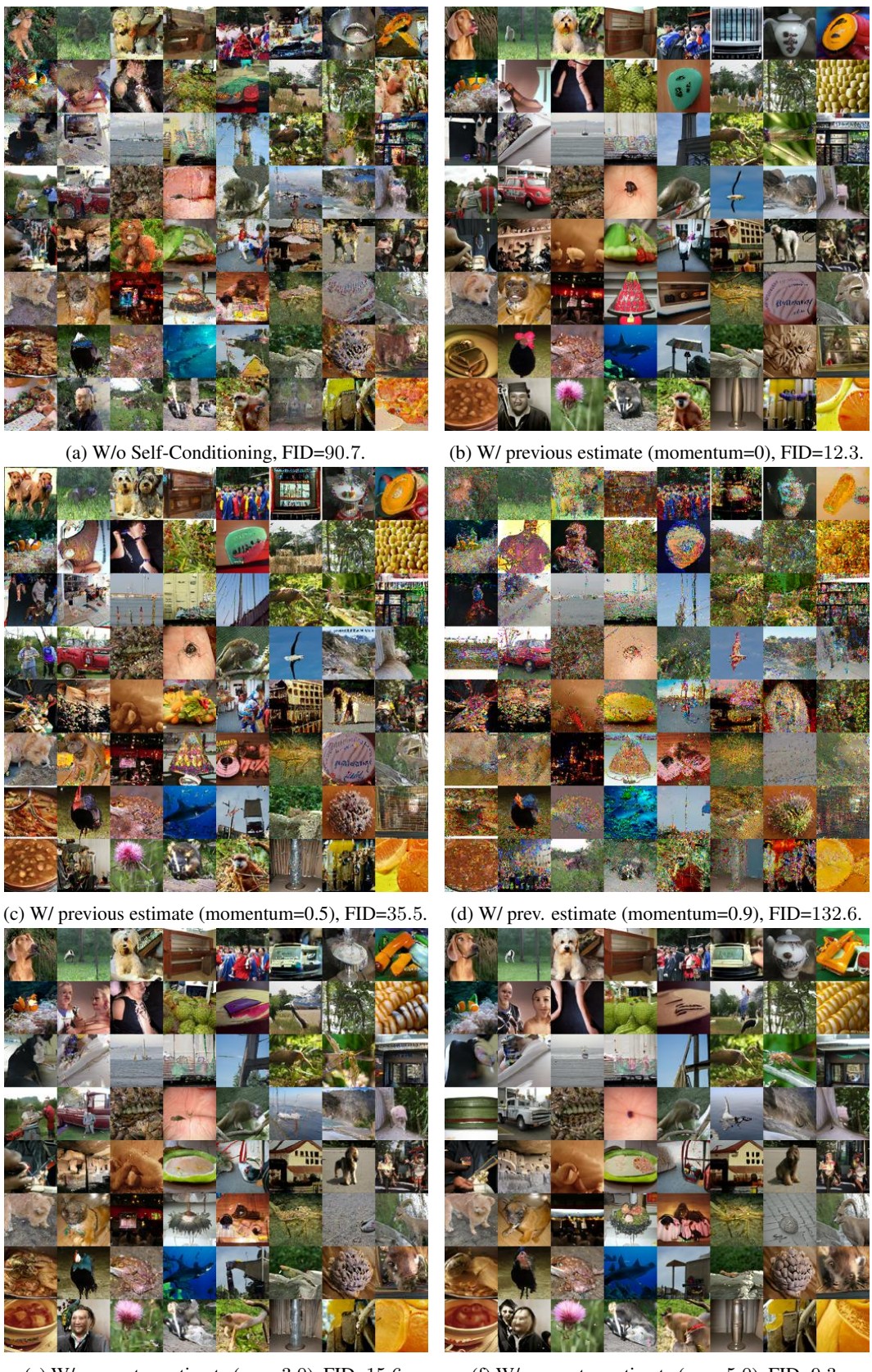

(a) W/o Self-Conditioning, FID=90.7.    (b) W/ previous estimate (momentum=0), FID=12.3.

(c) W/ previous estimate (momentum=0.5), FID=35.5.  (d) W/ prev. estimate (momentum=0.9), FID=132.6.

(e) W/ same-step estimate ($w = 3.0$), FID=15.6.  (f) W/ same-step estimate ($w = 5.0$), FID=9.3.

Figure 14: Random samples of Bit Diffusion with UINT8 (RAND) on categorical IMAGENET $64{\times}64$ using various Self-Conditioning sampling strategies. Different plots share the same set of $\boldsymbol{x}_T$. Sampling with 100 steps of DDIM without asymmetric time intervals.

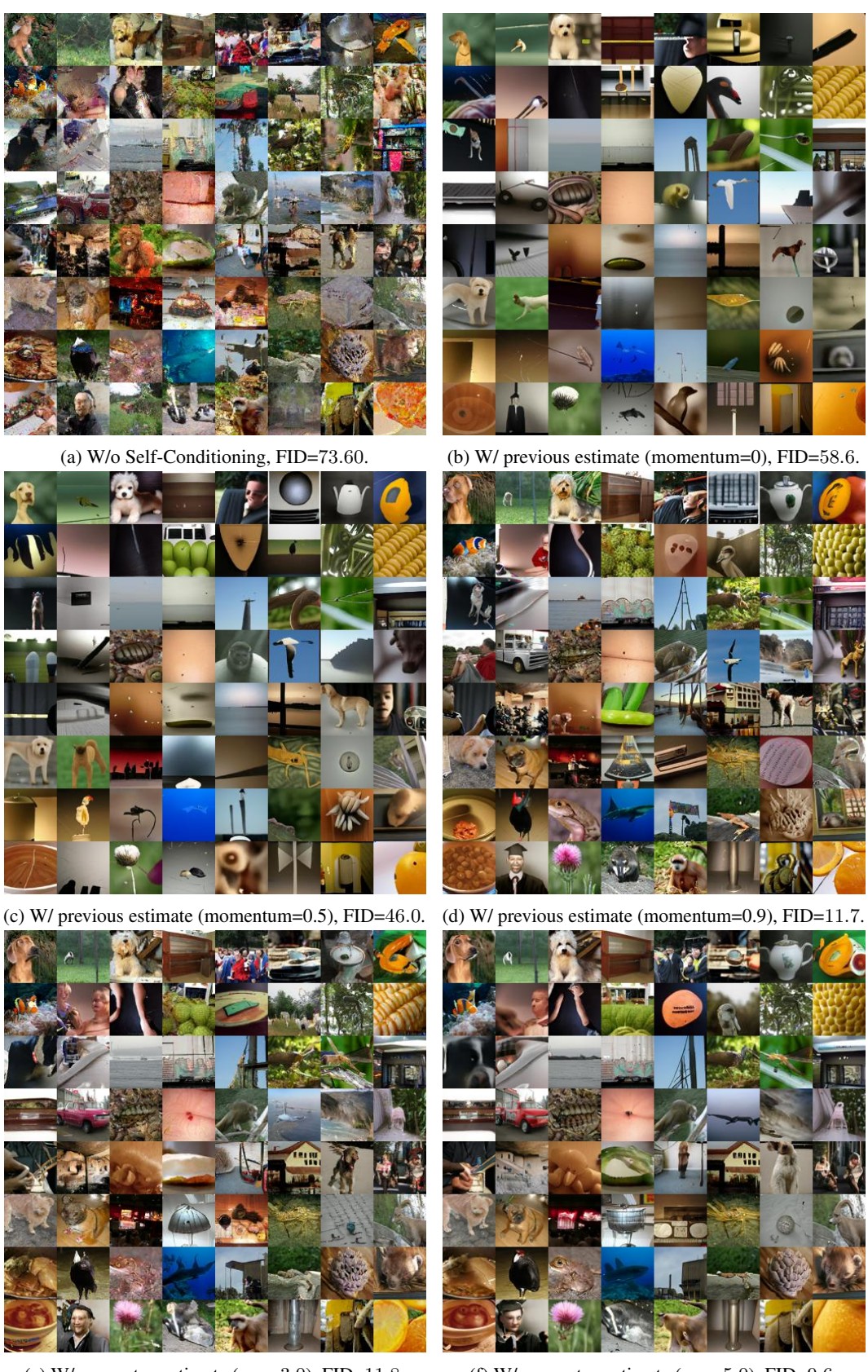

(a) W/o Self-Conditioning, FID=73.60.  (b) W/ previous estimate (momentum=0), FID=58.6.

(c) W/ previous estimate (momentum=0.5), FID=46.0.  (d) W/ previous estimate (momentum=0.9), FID=11.7.

(e) W/ same-step estimate ($w = 3.0$), FID=11.8.  (f) W/ same-step estimate ($w = 5.0$), FID=9.6.

Figure 15: Random samples of Bit Diffusion with UINT8 (RAND) on categorical IMAGENET $64 \times 64$ using various Self-Conditioning sampling strategies. Different plots share the same set of $\boldsymbol{x}_T$. Sampling with 1000 steps of DDIM without asymmetric time intervals.

