# OpenReview forum: "Analog Bits: Generating Discrete Data using Diffusion Models with Self-Conditioning"
_ICLR.cc/2023/Conference — ICLR 2023 poster_

### Official Review · Reviewer_YYSB · 2022-10-17

**Confidence:** 4
**Clarity, Quality, Novelty And Reproducibility:** The paper is very clear and readable.…
**Correctness:** 3
**Technical Novelty And Significance:** 2
**Empirical Novelty And Significance:** 2
**Recommendation:** 6

**Strength And Weaknesses:**

Strength:
The author shows that modeling bits is a simple and effective method. This makes it possible to directly use continuous diffusion models to model discrete data such as texts. Although the self-conditioning technique takes some additional time, it looks very useful to improve the generation performance. This method works for both images and texts.


Weakness:

1. In Figure 6 (a), "Asymmetric Time Intervals" only improves the sample quality when the number of steps is small. When the number of steps is large, a "time difference" of zero performs the best. It needs to tune the "time difference" hyperparameter carefully given a specific number of steps. So I wonder would it be more convenient to directly use a fast sampling method, such as [1, 2], for a good sample quality with a small number steps?

2. In Figure 6 (b), using a large number of steps for "UNIT8 (RAND)" leads to a worse performance. While the author empirically fixes this issue in Appendix, it would be better to provide some formal or intuitive explanation.

3. Since the text is also categorical data, like the "UNIT8 (RAND)", would the text also meets the same problem that a large number of steps lead to a worse performance?

[1] Analytic-DPM: an Analytic Estimate of the Optimal Reverse Variance in Diffusion Probabilistic Models

[2] DPM-Solver: A Fast ODE Solver for Diffusion Probabilistic Model Sampling in Around 10 Steps

**Summary Of The Paper:**

The main idea of this paper is to convert data to bit, and then model these bits using continuous diffusion models. This paper proposes two additional techniques: the self-conditioning to improve the x0 prediction, and the asymmetric time to improve the sampling speed. There are experiments on both images and texts, which show the effectiveness of this method.

**Summary Of The Review:**

The method is simple and looks effective under some cases, and meanwhile I have some concerns on the "Asymmetric Time Intervals" and the sample quality for categorical data. My recommadation is marginally above the acceptance threshold.

---

> ### Author Response · Authors · 2022-11-19
> **Response to reviewer YYSB**
>
> We thank the reviewer for the feedback and suggestions. Please find our responses below.
>
> **It needs to tune the "time difference" hyperparameter carefully given a specific number of steps**
>
> We believe the extra cost brought by the time difference parameter is pretty minimal, compared to other hyperparameters (such as learning rate, or architecture choices), as it only appears in inference and does not affect training. So it is very cheap to test different values, and can be validated with just a few generated samples.
>
> **Since the text is also categorical data, like the "UNIT8 (RAND)", would the text also meets the same problem that a large number of steps lead to a worse performance?**
>
> We did not find that using a larger number of steps (up to 250) lead to degeneration of the samples.
>
>
> **I wonder would it be more convenient to directly use a fast sampling method, such as [1, 2], for a good sample quality with a small number steps?**
>
> We took reviewer's suggestion and explored our techniques with DPM-solver. Here are some preliminary results we obtained.
>
> We do find that DPM-sampler provides a boost to diffusion models based on analog bits, similar to what it is able to do for continuous data. This shows a potential of our model enjoying faster continuous sampler while other baselines (e.g., D3PM) may not be able to do due to their use of discrete states. Table below shows the FID scores of bit diffusion models on ImageNet-64x64 under different binary encoding schemes. We find that the DPM-Solver sampler is able to provide a significant reduction in function evaluations for bit diffusion on discrete/categorical data (with 30 NFEs it gets comparable FIDs as 100 NFEs of DDIM), similar to that in continuous diffusion models.
>
> | Samplers | UINT8 | Gray Code | UINT8 (RAND) |
> |:---:|:---:|:---:|:---:|
> | DDIM @ 1000 NFE| 5.51 | 5.14 | 58.45 |
> | DDPM @ 1000 NFE| 7.71 | 6.91 | 64.08 |
> | DDIM @ 400 NFE | 5.00 | 5.52 | 38.44 |
> | DDPM @ 400 NFE | 4.84 | 5.37 | 40.91 |
> | DDIM @ 100 NFE | 8.80  | 11.31 | 8.76 |
> | DDPM @ 100 NFE | 13.04 | 12.77 | 9.25 |
> | | | |
> | DPM-Solver @ 30 NFE | 7.85 | 9.64 | 10.39 |
> | DPM-Solver @ 50 NFE | 6.46 | 7.61 | 10.96 |
>
> Furthermore, we also find that self-conditioning continues to provide a boost with DPM-solver. For example, the table below shows FID scores of diffusion models on ImageNet 64x64 (continuous rgb values). And we find that the self-conditioning consistently improves the performance of DPM-Solver with fixed number of function evaluations.
>
> | Model |  DPM-Solver @ 20 NFE | DPM-Solver @ 30 NFE |
> |:---:|:---:|:---:|
> | $\epsilon$ prediction, wo/ self-conditioning | 6.10 | 5.58 |
> | $\epsilon$ prediction, w/ self-conditioning | 4.24 | 4.15|
> | | | |
> | $x_0$ prediction, wo/ self-conditioning | 12.13 | 11.05 |
> | $x_0$ prediction, w/ self-conditioning | 6.94 | 6.43 |
>
> We will include the discussion in future revision.

---

> > ### Comment · Reviewer_YYSB · 2022-11-20
> > **Thanks for the reply**
> >
> > Thanks for the reply. I keep the score.

---

> > > ### Author Response · Authors · 2022-11-20
> > > **quick follow up**
> > >
> > > Thanks for your reply. We believe our responses should address reviewer's concerns expressed in the original review, but it does not convince the reviewer to change the overall evaluation/score of the work. We would appreciate if the reviewer can elaborate a bit on how our work could have been done differently, or what aspects should be improved, so that the reviewer may think it is better than "marginally above the acceptance threshold"?

---

> > > > ### Comment · Reviewer_YYSB · 2022-11-27
> > > > **Thanks for the reply**
> > > >
> > > > The second question is not fully addressed. This is really counter-intuitive. I think this phenomenon deserves a systematical study. Why does it occur for UNIT8 (RAND) on image? Why does it not occur for text data?

---

> > > > > ### Author Response · Authors · 2022-11-29
> > > > > **Thanks for the follow-up**
> > > > >
> > > > > We believe this is related to self-conditioning. As without self-conditioning, this does not seem to happen (but the quality is also much worse). We only observe it on images encoded as UINT8 (RAND), but NOT when applying self-conditioning to diffusion models for continuous images (rgb pixels), or images with UINT8 or GrayCode. We suspect this may have something to do with the model capacity, as 64x64 image generation with UINT8 (RAND) is the most difficult case (with 12K discrete tokens whose binary codes are fully shuffled). We agree this phenomenon remains mysterious and not fully explained. Feel free to let us know if the reviewer has suggestions on how this can be better probed. We believe this question belongs to the understanding of the proposed self-conditioning technique and its effectiveness, and we fully agree with the reviewer that there should be a systematic study of it. Just like many deep learning understanding papers studying the effectiveness and mysteries of many deep learning techniques, this one that we’re facing (with self-conditioning) may also deserve to be looked at in future understanding papers (since self-conditioning is very effective in generating both continuous and discrete data).

---

### Official Review · Reviewer_j3eh · 2022-10-23

**Confidence:** 4
**Correctness:** 3
**Technical Novelty And Significance:** 3
**Empirical Novelty And Significance:** 3
**Recommendation:** 8

**Clarity, Quality, Novelty And Reproducibility:**

The paper is well written and easy to follow. The paper is presenting an idea that the generation tasks can be modeled in bits level with diffusion models. However, the proposed method is not verified on large scale making it unclear on the impacts of those tricks in large scale.

**Strength And Weaknesses:**

Strength:
1. The paper is well written and easy to follow.
2. The proposed methods are simple yet effective and supported by the experiments.
3. The idea behind the paper is novel: modeling generative tasks in bits level with diffusion models.
4. The proposed algorithm is well presented and the is reproducible.

Weakness:
1. The proposed algorithm is not verified on larger dataset such as LAION-5B wit larger models. It is hard to judge if the proposed method still work when scaled to large dataset.
2. The two proposed tricks are working well on DDPM sampler. However, recently, a variety of efficient samplers are proposed such as DPM-Solver and PLMS sampler. Does the proposed tricks work on those efficient samplers?

**Summary Of The Paper:**

The paper proposes a novel idea of generating images in bits level. The proposed method is shown to be effective on selected dataset. The selected datasets are not simple and includes complex dataset such as COCO(in contrast, some of the works only report results on CIFAR and ImageNet, which are simpler than COCO). The two proposed tricks: 1. self-conditioning and 2 asymmetric time intervals are simple yet shown to be effective also. Overall, I like the paper.

**Summary Of The Review:**

Overall, I like the idea behind the paper. Although the proposed methods is not verified in large scale, it shows some promising results on complex tasks already (COCO dataset).

---

> ### Author Response · Authors · 2022-11-19
> **Response to reviewer j3eh**
>
> We thank the reviewer for the feedback and suggestions. Please find our responses below.
>
> **The proposed algorithm is not verified on larger dataset such as LAION-5B wit larger models. It is hard to judge if the proposed method still work when scaled to large dataset.**
>
> We agree it is an important question to ask how well the model scales up, but it also requires a non-trivial amount of work and resources to properly answer this question. In this work, we focus on developing and demonstrating this new promising approach at diverse modalities of images and languages, which lays a good foundation for the future attempt of scaling it up to larger datasets.
>
> **The two proposed tricks are working well on DDPM sampler. However, recently, a variety of efficient samplers are proposed such as DPM-Solver and PLMS sampler. Does the proposed tricks work on those efficient samplers?**
>
> We follow the reviewer’s suggestion in exploring our techniques with DPM-solver. Here are some preliminary results we obtained.
>
> Firstly, we find that self-conditioning continues to provide a boost with DPM-solver. For example, the table below shows FID scores of diffusion models on ImageNet 64x64 (continuous rgb values). And we find that the self-conditioning consistently improves the performance of DPM-Solver with fixed number of function evaluations.
>
> | Model |  DPM-Solver @ 20 NFE | DPM-Solver @ 30 NFE |
> |:---:|:---:|:---:|
> | $\epsilon$ prediction, wo/ self-conditioning | 6.10 | 5.58 |
> | $\epsilon$ prediction, w/ self-conditioning | 4.24 | 4.15|
> | | | |
> | $x_0$ prediction, wo/ self-conditioning | 12.13 | 11.05 |
> | $x_0$ prediction, w/ self-conditioning | 6.94 | 6.43 |
>
> Secondly, we find that DPM-sampler also provides a boost to diffusion models based on analog bits, similar to what it is able to do for continuous data. This shows a potential of our model enjoying faster continuous sampler while other baselines (e.g., D3PM) may not be able to do due to their use of discrete states. Table below shows the FID scores of bit diffusion models on ImageNet-64x64 under different binary encoding schemes. We find that the DPM-Solver sampler is able to provide a significant reduction in function evaluations for bit diffusion on discrete/categorical data (with 30 NFEs it gets comparable FIDs as 100 NFEs of DDIM), similar to that in continuous diffusion models.
>
> | Samplers | UINT8 | Gray Code | UINT8 (RAND) |
> |:---:|:---:|:---:|:---:|
> | DDIM @ 1000 NFE| 5.51 | 5.14 | 58.45 |
> | DDPM @ 1000 NFE| 7.71 | 6.91 | 64.08 |
> | DDIM @ 400 NFE | 5.00 | 5.52 | 38.44 |
> | DDPM @ 400 NFE | 4.84 | 5.37 | 40.91 |
> | DDIM @ 100 NFE | 8.80  | 11.31 | 8.76 |
> | DDPM @ 100 NFE | 13.04 | 12.77 | 9.25 |
> | | | |
> | DPM-Solver @ 30 NFE | 7.85 | 9.64 | 10.39 |
> | DPM-Solver @ 50 NFE | 6.46 | 7.61 | 10.96 |
>
> We will include the discussion in future revision.

---

### Official Review · Reviewer_h3v1 · 2022-10-24

**Confidence:** 4
**Correctness:** 4
**Technical Novelty And Significance:** 3
**Empirical Novelty And Significance:** 3
**Recommendation:** 8

**Clarity, Quality, Novelty And Reproducibility:**

* The paper is well-written with clear exposition of ideas.
* Enough details are provided to enable reproducibility.

**Strength And Weaknesses:**

Strengths:
* The idea of treating discrete data as continuous bits, is simple yet effective.
* The proposed self-conditioning and asymmetric time intervals techniques seem to improve the performance significantly and are also applicable to other continuous diffusion models.

Weaknesses:
* The authors should discuss we the experiments have been over small size images (only 64 by 64). If the computational cost is the main concern, it should be delineated if the expectation is that the same performance will be observed for higher resolutions or not.
* Harder constraint or regularization to enforce the binary outputs has not been adopted based on the assumption that the model expressivity will take care of it. However, conducting experiments to show additional regularization won't benefit the performance will be valuable.

**Summary Of The Paper:**

This paper proposes the simple yet effective idea of using continuous diffusion models to generate discrete data by representing them as binary data and modeling these binary data as real numbers. The authors also propose two techniques, namely Self-Conditioning and Asymmetric Time Intervals, which lead to improvement in sample quality.

**Summary Of The Review:**

Overall, the main idea of this paper is novel and simple. The experiments show the good performance of the analog bits idea, and this work paves the way for future works on generating discrete data using continuous diffusion models.

---

> ### Author Response · Authors · 2022-11-19
> **Response to Reviewer h3v1**
>
> We thank the reviewer for their feedback.  We address the questions as follows.
>
> **The authors should discuss we the experiments have been over small size images (only 64 by 64).**
>
> We follow existing baselines (such as D3PM, τLDR-10, autoregressive model) in setting up the discrete/categorical image generation tasks. In fact, all these methods only have evaluations on CIFAR-10. We have gone above and beyond to test it and report metrics on ImageNet-64x64 (for which our baselines have not reported any numbers, due to scalability and other reasons). Furthermore, even for continuous diffusion models trained on ImageNet, people need to use techniques like cascading / multi-stage / super-resolution / classifier guidance to train good models for images larger than 64 x 64, so it becomes orthogonal to the core of our contributions (especially without any existing baselines on discrete images of larger sizes to compare with).
>
> **Harder constraint or regularization to enforce the binary outputs has not been adopted based on the assumption that the model expressivity will take care of it. However, conducting experiments to show additional regularization won't benefit the performance will be valuable.**
>
> During the inference, we did clipping of the x_0 prediction to make sure they are within -1 and 1 (similar tricks are used for continuous diffusion models on real-valued pixels), but this seems to have minimal effect for most cases. We also tried using a tanh activation at the output of the denoising network so that prediction would be naturally bounded, but again that doesn’t seem to help. Therefore, the model doesn’t seem to rely on hard constraints to work well.

---

### Official Review · Reviewer_ECCX · 2022-10-24

**Confidence:** 4
**Correctness:** 4
**Technical Novelty And Significance:** 4
**Empirical Novelty And Significance:** 4
**Recommendation:** 6

**Clarity, Quality, Novelty And Reproducibility:**

The method is clearly explained and the necessary ablations have been conducted.

**Strength And Weaknesses:**

Strengths:
* The authors introduce two techniques - self-conditioning and asymmetric time intervals. Both of these seem useful for both continuous and discrete diffusion models. Other concurrent work has also shown that using self-conditioning improves results of diffusion models.
* Figure 7 shows that the generated bits are concentrated on two modes without adding explicit constraints and this validates the authors' hypothesis that modeling binary bits is easy.
* For image captioning, the AnalogBits achieves the same performance as auto-regressive models with just 10 diffusion steps!

Weaknesses:
* Discrete diffusion models are especially suitable for NLP tasks given the discrete nature of text. Despite, proposing a method for discrete data, the authors only presents results on one NLP task (image captioning). Additional experimental results on other NLP texts would help strengthen this paper.
* For encoding NLP tokens as analog bits, each token is mapped to a random set of 15 analog bits. As a results, there is no semantic meaning in the new representation (similar tokens don't have similar bit representations). The authors show that using UINT8 is much better when compared to using random bits for discrete image generation. Exploring ways to represent semantic meaning in bit representations of tokens would be an interesting extension to the presented work.
* Gray codes perform worse than UINT8 and no explanation is give as to why this is the case.

**Summary Of The Paper:**

The authors of this paper introduce a method to use diffusion models to generate discrete data. The crux of the method rests on the observation that discrete data can be represented with binary bits and binary bits can be uses as real numbers to train a continuous diffusion model; the output at the end of the diffusion process can be rounded to obtain binary output bits.

**Summary Of The Review:**

Diffusion models have shown strong performance in continuous domains. However, there are many domains that are not continuous (like text) and adapting diffusion models to discrete settings will aid in exploring whether state-of-the-art results can be improved by using diffusion models for discrete domains. The method in this paper also converts discrete domain data to a "continuous representation" and therefore additional work on continuous diffusion models can easily be ported and used for the discrete setting. Furthermore, self-conditioning and asymmetric time intervals are useful general techniques that will be of interest to the broader diffusion model community.

---

> ### Author Response · Authors · 2022-11-19
> **Response to Reviewer ECCX**
>
> We thank the reviewer for the feedback and suggestions.  Below, please find our responses to each of the concerns or questions raised in the review:
>
> **The authors only presents results on one NLP task (image captioning). Additional experimental results on other NLP texts would help strengthen this paper.**
>
> We agree that our model are suitable for broader NLP tasks (beyond image captioning). However, due to the limited resources, we have to choose either many NLP tasks or a mix of image and language tasks. Given that many baselines we compare to are developed and evaluated with discrete/categorical image data, and the diversity of the two modalities over one modality, we decided to do a mix of image and language tasks, rather than many NLP tasks. We believe, in our work, sufficient empirical evidence has been provided to demonstrate our approach is a promising new method in modeling high dimensional discrete data, for both domains of images and languages. While it is always beneficial to test and apply a new approach to more tasks, it should also progress with time. For example, a lot of applications of Transformers (such as BERT, GPT) appeared years after the Transformers were introduced. We hope similar cases can be made for our work.
>
> **Exploring ways to represent semantic meaning in bit representations of tokens would be an interesting extension to the presented work.**
>
> We did find that different binary encoding (uint8 vs uint8 shuffle as an example) has an effect on the final results, so one would expect that learned binary codes may be a better representation of data for training the diffusion models. So we agree it would be an interesting and promising direction to explore in the future.
>
> **Gray codes perform worse than UINT8 and no explanation is give as to why this is the case.**
>
> Gray codes are indeed worse than UINT8 in terms of FID/IS metrics, but only slightly. We expect the reason is related to that in gray code, pixel intensity values (integers) in a wide range only differ by 1 bit in their binary codes, as demonstrated in Figure 8 of the paper. Therefore, when the model mis-predicts a single bit, it could result in very different pixel intensity.

---

> > ### Comment · Reviewer_ECCX · 2022-11-27
> > **Response to authors**
> >
> > Thanks for the response. I still think presenting results on language modeling would improve the quality of the paper and would make comparisons against other diffusion models designed for language tasks (like Diffusion-LM [1]) possible.
> >
> > [1] Diffusion-LM Improves Controllable Text Generation

---

> > > ### Author Response · Authors · 2022-11-29
> > > **Thanks for the following-up**
> > >
> > > We thank the reviewer for the suggestion, and we agree that applying our approach to large-scale language modeling tasks is an important future work. It is worth to note we already included an image captioning task which is a language modeling task (conditioned on images), and as a proof of concept, we show our approach works on par with autoregressive language modeling in terms of captioning metrics.
> > >
> > > The reasons why we do not expand the scope to include more large scale language modeling tasks are as follows. First, it is very challenging to evaluate diffusion models for language modeling tasks in a very convincing manner, especially on a small scale. For example, autoregressive LM uses perplexity, but for diffusion models, computing perplexity is not straight-forward and may not be accurate (eg only lower bound). It is also not clear if the sample quality & downstream performance are well correlated with the perplexity under diffusion models. Secondly, It is very difficult to show the value of diffusion LM given that generally language modeling and downstream tasks have short sequences (<1000), and autoregressive models can already run reasonably fast. If the value cannot be very clearly demonstrated, it is just another proof of concept as we have already done with image captioning. Finally, the power of language models is in scale, meaning small language models are not nearly as useful and conclusions derived from them often do not transfer to large scales. Large scale training (eg GPT scale) is rarely done and it wouldn’t be reasonable to ask a novel methodology paper (with 3 new techniques proposed) to also include that.
> > >
> > > Therefore, our work aims to propose new methods, and provided solid proof-of-concept experiments in both image and text modalities to show this is a very promising approach (beating other SOTAs). Whether it is going to be a mainstream approach complementing/replacing autoregressive language model, and/or being a game changer to the field of discrete data generation, it has to be evaluated over time by many subsequent work.

---

> ### Comment · Area_Chair_npUA · 2022-11-23
> **inconsistent ratings between "correctness" and "recommendation"**
>
> Dear Reviewer ECCX,
>
> Your rated recommendation as 6 but correctness as "1: The main claims of the paper are incorrect or not at all supported by theory or empirical results." Would you please confirm whether this was a typo or whether you indeed had serious concerns on the correctness of the paper?
>
> Thanks,
>
> AC

---

> > ### Comment · Reviewer_ECCX · 2022-11-23
> > **Response to Area Chair**
> >
> > Yes, this was indeed a typo. I've fixed it.

---

### Comment · Area_Chair_npUA · 2022-11-23
**Missing discussion on computational complexity**

Dear authors,

Reviewer h3v1 had indicated some concern on computational cost, which may limit the size of the images, but your response had not directly addressed that. It seems that for a 64 $\times$ 64 $\times$ 3 RGB image under 8bit binary encoding, the latent dimension of the image with analog bits would become 64 $\times$ 64  $\times$ 8 $\times$ 3? How would that impact the computational complexity? More specifically, how much more costly than DDPM/D3PM under the same number of diffusion steps in both training and generation?

Thanks,

AC

---

> ### Author Response · Authors · 2022-11-23
> **Clarification**
>
> Dear AC,
>
> Thanks for bringing this up, as we may have mis-interpreted the original question.
>
> You're correct that with analog bits, an image of size H x W x 3 would become H x W x 24. This increases the memory for storing image tensor and computation for the first/last convolution layer as the first/last hidden tensor will be H' x W' x d where d is hidden dimension (e.g. 256).  Compared to the majority of network computation (mostly convolutions on channels of 256 or larger), we believe this increase in first/last layer computation is relatively small/negligible.
>
> Now if we look at D3PM where there are 256 discrete states for each subpixel, so the new image size would be H x W x 768 (due to 256*3), this consumes more than an order of magnitude more memory as well as computation for the first/last convolution layers compared to analog bits, which makes it difficult to scale to images larger than 32x32 (D3PM didn't have ImageNet 64x64 results).
>
> Finally, DDPM considers images as continuous data, which is a different experimental setting to D3PM and ours, where images are treated as discrete/categorical data. We acknowledge that treating image as discrete/categorical data has limited practical use (eg may be useful for compression), but undeniably it is a very challenging benchmark for modeling high-dimensional discrete/categorical data. Compared to other discrete data (e.g. text), image generation is relatively easy to evaluate at small / medium scale (due to the reasonable correlation between FID/IS and visual perception). So the fact that many discrete diffusion models / autoregressive models use such benchmark, and we beat them by a large margin indicates how good / efficient our method is (in modeling high dimensional discrete/categorical data).
>
> Thanks,
> Authors

---

> > ### Comment · Area_Chair_npUA · 2022-11-23
> > **Re: Clarification**
> >
> > Dear authors,
> >
> > Thanks for your clarification. Would you be able to add some specific numbers on the actual time comparison on a specific GPU between $H \times W \times 3$,  $H \times W \times 24$, and  $H \times W \times 768$?  That would help the readers to have a better idea on time complexity.
> >
> > Your response "Finally, DDPPM..." about handling discrete data makes sense. The experiments on images show that encoding continuous data with 8-bit binary encoding, and modeling each binary bit as Gaussian in the latent space, is a sound choice. However, this does not necessarily mean the proposed procedure is a sound solution for data that occur naturally in a discrete format, which could be extremely sparse (significantly more zero than non-zeros in a binary/count setting; implying label imbalance issues) and over-dispersed (variance larger than mean; in a naive binary encoding scheme, this could imply lots of binary codes will rarely be observed; e.g., in bag-of-words, the word count follows the zipf law) and hence may not be well modeled in a continuous/Gaussian latent space.
> >
> > While there are few baselines in diffusion-based models who have carefully benchmarked their performance on naturally-occurring discrete data, such as bag-of-words vectors in text analysis and movie-user rating matrix in recommendation system, there are plenty of non-deep-generative-model-based algorithms, e.g., traditional shallow Poisson factor analysis/topic models for text analysis and their deep-learning versions, that are targeting that type of data.
> >
> > With that said, I am willing to accept the claim "So the fact that many discrete diffusion models / autoregressive models use such benchmark, and we beat them by a large margin indicates how good / efficient our method is (in modeling high dimensional discrete/categorical data)." But I hope the authors could clarify the potential limitation of analog bits in modeling naturally-occurring discrete data that are often extremely sparse and overdispersed.
> >
> > Thanks,
> >
> > AC

---

> > > ### Author Response · Authors · 2022-11-29
> > > **Thanks for following-up**
> > >
> > > We agree discrete tokens from natural images have different distributions as discrete tokens in text. That is why we tested our method in both discrete/categorical image generation and text generation (image captioning task) to better demonstrate the effectiveness of the approach compared to previous SOTAs.
> > >
> > > For actual time comparison of different choices of image encoding. Here’s a breakdown of forward pass time of U-Net (used in ImageNet-64x64) on a single V100 GPU, with batch size of 32 for different encodings of 64x64 images:
> > >
> > > | Encoding        | Time|
> > > |--------------------------------|-------|
> > > | 64x64x3 (rgb encoding)         | 367ms |
> > > | 64x64x24 (analog bit encoding) | 370ms |
> > > | 64x64x768 (D3PM encoding)      | 396ms |
> > >
> > > Numbers are averaged over 100 runs for each.

---

### Decision · Program_Chairs · 2023-01-20

**Decision:**

Accept: poster

**Justification For Why Not Higher Score:**

There are some concerns on the level of novelty and the limitation on the proposed approach on naturally-occurred discrete data.

**Justification For Why Not Lower Score:**

The empirical performance on the chosen tasks is convincing. The paper provides a valuable addition to the application of diffusion models.

**Metareview: Summary, Strengths And Weaknesses:**

This paper proposes a simple but effective method to extend diffusion models to generate discrete data. The basic idea is to encode a discrete data with a vector of binary variables, which are linked to latent continuous variables that are modeled with continuous diffusion models. Two techniques, including Self-Conditioning and Asymmetric Time Intervals, have also been introduced to improve the underlying diffusion models. While there are some concerns on the level of novelty and the limitation on the proposed approach on naturally-occurred discrete data, all reviewers recognize the contributions in providing a simple and effective solution of generating discrete data in two different types of tasks and are in favor of accepting the paper.

**Note From Pc:**

if the above contains the word "oral" or "spotlight" please see: "oral" presentation means -> notable-top-5% and "spotlight" means -> notable-top-25%. As stated in our emails, we are disassociating presentation type from AC recommendations